# The silkworm (*Bombyx mori*) gut microbiota is involved in metabolic detoxification by glucosylation of plant toxins

Shuangzhi Yuan[1], Yong Sun[1], Wenqiang Chang[1], Jiaozhen Zhang[1], Jifa Sang[2], Jiachun Zhao[2], Minghui Song[1], Yanan Qiao[1], Chunyang Zhang[1], Mingzhu Zhu[1], Yajie Tang[3] & Hongxiang Lou [1✉]

Herbivores have evolved the ability to detoxify feed components through different mechanisms. The oligophagous silkworm feeds on *Cudrania tricuspidata* leaves (CTLs) instead of mulberry leaves for the purpose of producing special, high-quality silk. However, CTL-fed silkworms are found to have smaller bodies, slower growth and lower silk production than those fed mulberry leaves. Here, we show that the high content of prenylated isoflavones (PIFs) that occurred in CTLs is converted into glycosylated derivatives (GPIFs) in silkworm faeces through the silkworm gut microbiota, and this biotransformation is the key process in PIFs detoxification because GPIFs are found to be much less toxic, as revealed both in vitro and in vivo. Additionally, adding *Bacillus subtilis* as a probiotic to remodel the gut microbiota could beneficially promote silkworm growth and development. Consequently, this study provides meaningful guidance for silk production by improving the adaptability of CTL-fed silkworms.

[1] Department of Natural Products Chemistry, Key Laboratory of Chemical Biology of the Ministry of Education, School of Pharmaceutical Sciences, Shandong University, Jinan 250012, P. R. China. [2] Linyi University, Yishui, Linyi 276400, P. R. China. [3] State Key Laboratory of Microbial Technology, Shandong University, Qingdao 266237, P. R. China. ✉email: louhongxiang@sdu.edu.cn

In nature, plants defend themselves from herbivores by producing toxic metabolites, while herbivores evolved the mechanisms to resist plant defences in adaptation to overcoming toxic feeding[1] by metabolic detoxification, including the destruction[2], hydrolysis[3], phosphorylation and glycosylation[4] of the toxic components. The above detoxification events were initiated by insect ATP-Binding Cassette transporters[5], intestinal microbes[6,7], or even horizontal gene transfer[8,9]. Among these factors, the intestinal microbiota plays important roles in defence and protection for insects[7,10]. For instance, pinewood nematode or pine weevil are insect herbivores reared on conifer forests rich in toxic terpenoids, such as α-pinene and diterpene acid, whose gut microbiota exhibit a strong ability to degrade terpenoids to contribute to insect fitness[11,12]. It was revealed that the beneficial interactions between insects and their gut microbiota achieve detoxification for hosts.

The silkworm, *Bombyx mori*, belonging to Lepidoptera, Bombycidae, as one of the oldest economic insects for silk production, has been widely cultured in the long history of sericulture in China[13]. As an oligophagous insect, silkworms mainly feed on mulberry leaves belonging to the Moraceae plant family but can also feed on *Cudrania tricuspidata* leaves (CTLs)[14] of the same family. It is recorded that CTL-fed silkworms have a long history and have been traced back to the Chinese ancient lexicon "Erhya". Notably, the cultivated silk produced by silkworms fed on CTLs, compared with mulberry or tussah silk, was much tougher with a stable structure, stronger tensile strength and better performance; especially suitable for making strings or bowstrings[15,16]; and sometimes used as a special material for making Dragon Robes in the old dynasty mentioned in "Chinese Technology in the Seventeenth Century: T'ien-kung k'ai-wu". Thus, CTLs have gradually become an alternative to feed silkworms in different production practices.

However, silkworms fed on CTLs with less adaptability easily had smaller bodies, slower growth[17,18] and lower silk production than those fed other leaves. The underlying causes of this phenomenon are unknown, although it has been reported that the lack of adaptation may be related to the secondary metabolites in CTLs for the observation of the upregulation of carboxylesterase, a detoxifying metabolic enzyme whose activity is regulated by related secondary metabolites[19]. Carboxylesterase activity in silkworms fed CTLs was higher than that in silkworms fed mulberry leaves, which suggested that toxic secondary metabolites existed in CTLs[18].

Here, by comparative chemical investigation of CTLs and silkworm faeces (SWFs), we found that prenylated isoflavones (PIFs), the principal constituents in CTLs, were converted to glycosylated derivatives (GPIFs) in SWFs, and the toxicity of GPIFs was greatly attenuated. This conversion was confirmed by the coculture test of 6,8-diprenylorobol (DPL), the main component of CTLs with silkworm intestinal microbes in vitro. Adding *B. subtilis* as a silkworm intestinal probiotic during feeding can remodel the intestinal microbiota as measured with 16S rDNA amplicon sequencing, and the growth and development of CTL-fed silkworms were well improved. Our research revealed the underlying mechanism of silkworm growth differences with different feeding materials. We also provide a useful way to improve the development of silkworms by remodelling the gut microbiota by adding probiotics during feeding.

## Results

**Naturally occurring GPIFs isolated from SWFs.** To explore these PIFs in CTLs with an unexposed influence on silkworms, the chemical components in SWFs produced by silkworms fed on CTLs were systematically isolated and purified. First, we obtained a total of 33 compounds from SWFs (Fig. 1), mainly PIFs and GPIFs, which constituted the major components in SWFs. In addition to 10 known PIFs, 6,8-diprenylorobol(**24**)[20], lupalbigenin (**25**)[21], isolupalbigenin (**26**)[22], auriculasin (**27**)[23], 4′-O-methylerythrinin C (**28**)[24], lupiwighteone (**29**)[25], erysenegalensein E (**30**)[26], millewanins H (**31**)[27], isoerysenegalensein E (**32**)[28] and millewanins G (**33**)[27], regarded as the main secondary metabolites in CTLs, 23 GPIFs were also obtained (**1-23**), including 21 GPIFs not described before to the best of our knowledge, silexcrins A-U (**1-21**), and two known ones, lupiwighteone 7-β-D-glucoside (**22**)[29] and genisteone (**23**)[30], which are *O*-glycosylated derivatives mainly introducing one or two glucoses into 7,3′,4′-hydroxyls at the prenylated isoflavone skeleton. In addition, two trivial pairs of epimers **13-16** were also obtained and identified with 2″*R*, 2″*S*, 3‴′*S*, and 3‴′*R* carbon stereo-centres by calculated ECD (Fig. 2). The structural elucidation of compounds **1-21** can be seen in Supplementary Notes 1 and 2, Supplementary Tables 1–8, and Supplementary Figs. 1–214 for details.

**Comparative chemical profile analysis of SWFs and CTLs.** Given that the characteristic constituents from SWFs and CTLs vary greatly, the main differences in components between SWFs and CTLs were limited to the peaks with retention times of 19–28 min in HPLC qualitative analysis (Fig. 3a). It included GPIFs as the major differential constituents. GPIFs **1-5, 6, 9** and DPL (**24**) were the principal components in SWFs; however, only DPL (**24**) was the major component in CTLs. In addition, based on the analysis of chemical structures, it was hardly surprising to find a structural correlation between **1-5** and **24**. Compounds **1-5** were formed by glucosylation of the aglycone DPL (**24**) (Fig. 3b).

Because GPIFs are largely abundant in SWFs, we preliminarily hypothesised that these GPIFs may be generated through gut metabolism in silkworm. Combined with the quantitative analysis of six principal components **1-5** and **24** in SWFs and CTLs by LC–MS (Fig. 3c–e), the contents of GPIFs **1, 2, 3, 4** and **5** in SWFs were 0.88, 2.67, 0.95, 1.17, and 1.61 mg/g, respectively, while they were not detected in CTLs. However, the content of DPL (**24**) in CTLs (12.41 mg/g) was approximately 6 times greater than that in SWFs (2.14 mg/g), which suggested that the reduced portions of DPL (**24**) in SWFs underwent metabolic transformation, partially involved in glycosylation conversion into GPIFs **1-5**.

It has demonstrated that those GPIFs were present in SWFs but not in CTLs. Next, we further studied the role of silkworm gut microbiota in intestinal metabolism for the formation of these GPIFs.

**Microbial biotransformation by intestinal microbes in vitro.** To further determine the influencing factors of glycosylation in the silkworm gut, we isolated the intestinal bacteria of silkworm and three isolates were obtained. We submitted the 16S rDNA sequences of these strains to NCBI for BLAST comparison, and found that the homology with *B. subtilis* SKG9 (Accession: OQ299533.1), *Staphylococcus sciuri* CTSP9 (Accession: EU855191.1) and *Enterobacter hormaechei* ECC33 (Accession: CP098486.1) were up to 99.87%, 99.87% and 99.93%, which suggested that these endogenous isolates were *B. subtilis*, *S. sciuri* and *E. hormaechei* strain, respectively.

It was also found that there existed *Bacillus* species, a probiotic group in intestinal flora with glycosyltransferases, which played an important role in glycosylation[31–35]. This kind of bacteria, as one of the important microbes in the silkworm gut, could influence the growth and development of silkworm, nutrition and metabolism[16,36]. To verify the glycosylation function of *Bacillus* species on PIFs, three *Bacillus* species, *Bacillus licheniformis* K1-

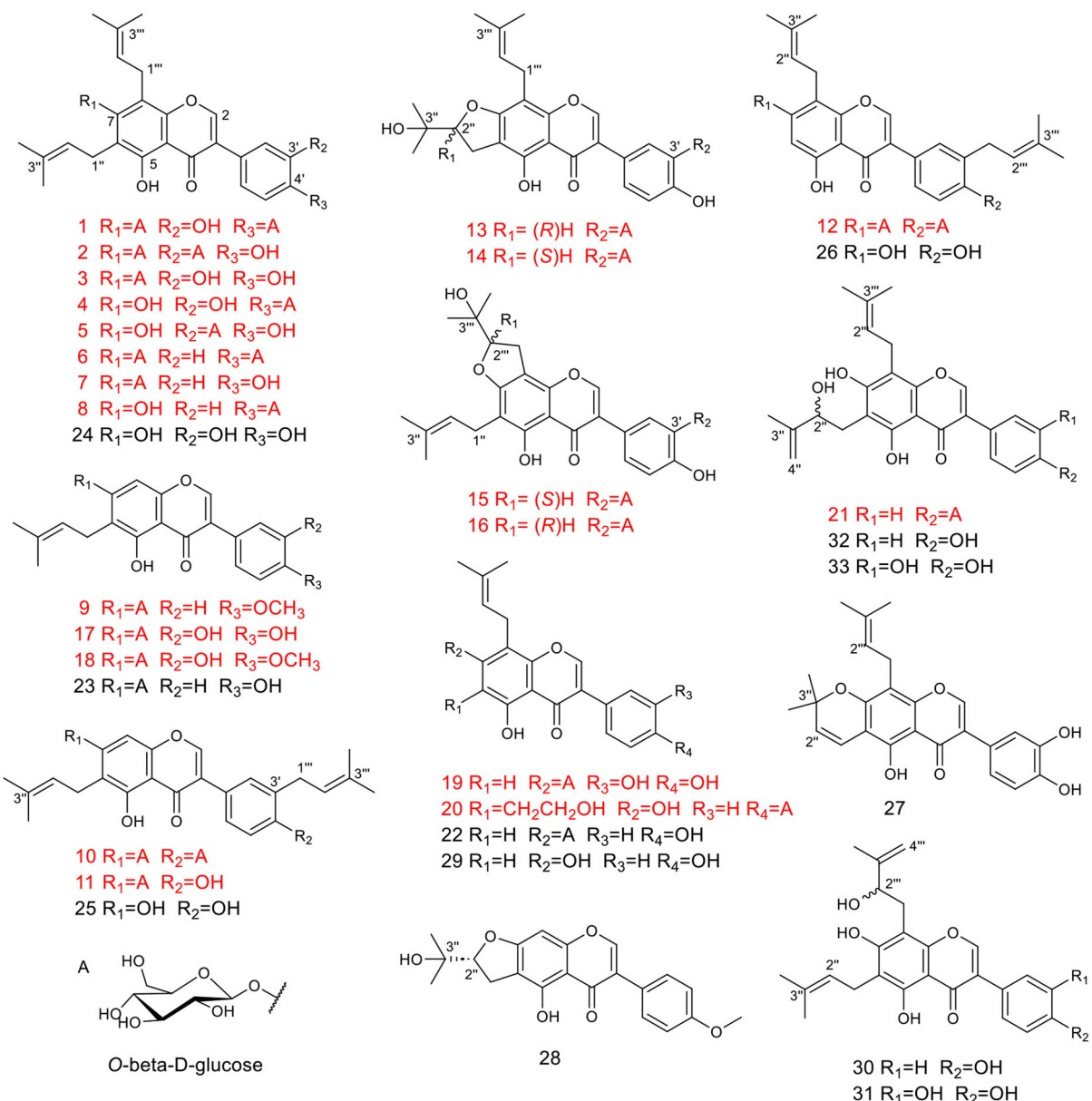

**Fig. 1 Naturally occurring PIFs and GPIFs isolated from SWFs.** Structures of compounds **1-33** are displayed. The new compounds **1-21** are shown in red.

30-2, *Bacillus licheniformis* K7-30-7 and the endogenous isolate, *B. subtilis* from silkworm gut, were selected to demonstrate their conversion capacity in vitro (Fig. 4a). According to the results displayed by LC–MS (Fig. 4b), the corresponding GPIFs **1-5** could be detected with success in experimental groups fermented with the three strains of *Bacillus* with DPL (**24**) as substrate. Therefore, all three *Bacillus* species were able to biotransform **24** into its glycosylated products.

Additionally, two pairs of epimers **13-16** were identified in SWFs, but no corresponding aglycones of **13-16** were found in CTLs, suggesting that compounds **13-16** were formed by microbiota transformation. This conclusion was further confirmed by the formation of **2** and **13-16** (Fig. 4c), which were converted by the three *Bacillus* species fed **5** in vitro and detected by LC–MS with the corresponding retention times and deprotonated molecular ion peaks (Supplementary Fig. 215). Here, we logically propose the microbial biotransformation of the principal component DPL (**24**) in SWFs by *Bacillus* species in the silkworm gut. First, DPL (**24**) was glycosylated by *Bacillus* species

in the silkworm gut to **3, 4** and **5**, and then sugars were successively added to form **1** and **2**. Moreover, a portion of **5** participated in the formation of **13-16** by successive epoxidation and SN2 nucleophilic addition under the influence of gut bacteria (Fig. 4d).

Moreover, we also conducted in vitro microbial transformation of DPL (**24**) by the other endogenous silkworm intestinal strains, *S. sciuri* and *E. hormaechei*. And *S. sciuri* could glycosylate DPL (**24**) to form **1, 2** and **5**, and *E. hormaechei* only converted substrate **24** to **2**, which inferred that these two isolates could only partially convert DPL (**24**) to produce corresponding GPIFs (Supplementary Fig. 216). Therefore, it was suggested that these silkworm intestinal isolates were involved in the conversion of PIFs, and *Bacillus* species showed better transformation ability of DPL (**24**) than the other two isolates from silkworm gut. To our regret, no other endogenous *Bacillus* from silkworm gut except *B. subtilis* were investigated, and we also cannot rule out the contribution of other microbial taxa in silkworm gut in the current study.

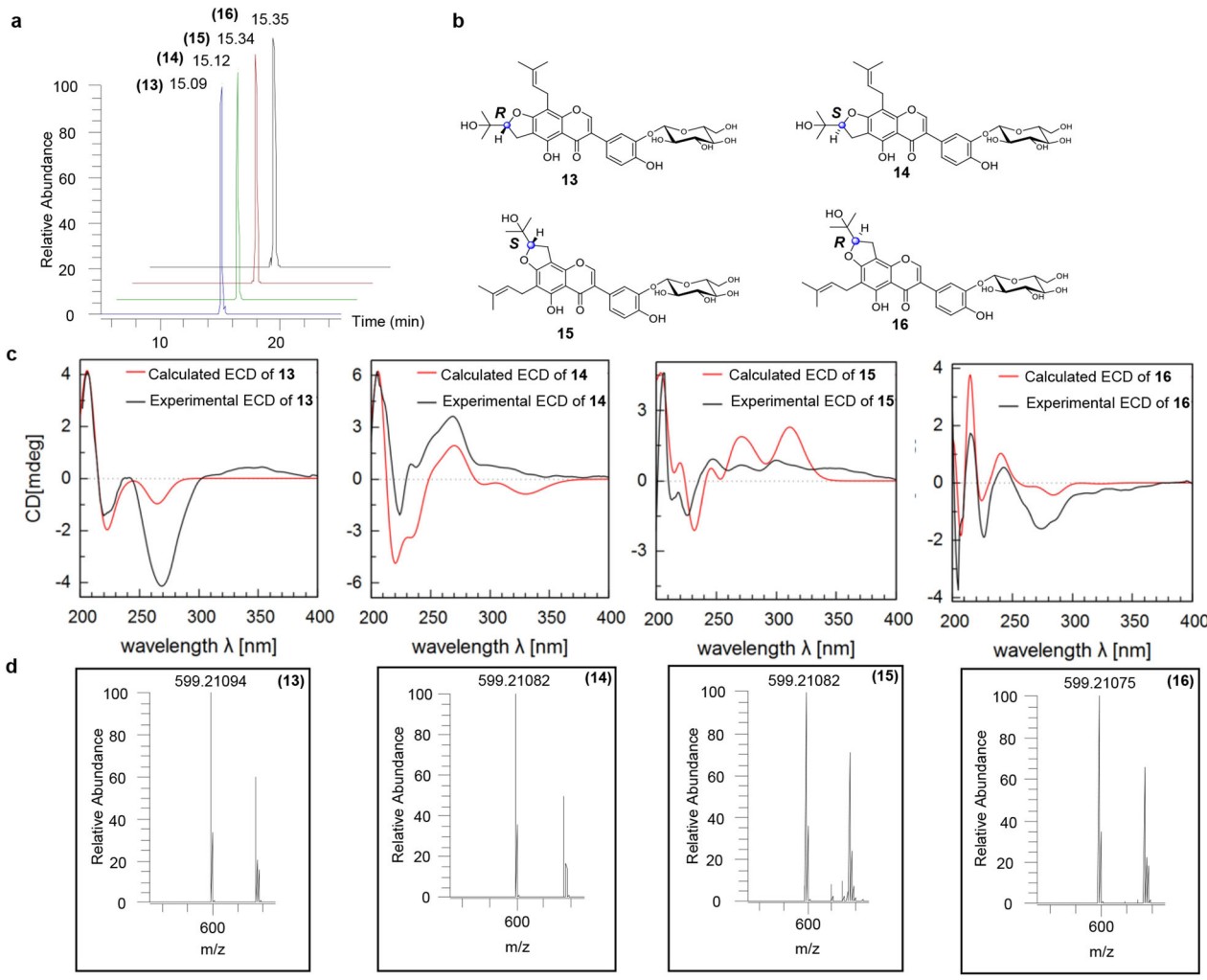

**Fig. 2 Determination of the absolute configuration of compounds 13-16. a** Base peaks of silexcrins M-P (**13-16**) under LC–MS analysis conditions. The retention times of compounds **13-16** were determined to be 15.09, 15.12, 15.34, and 15.35 min by LC–MS, respectively. **b** Structures of compounds **13-16**. **c** Corresponding experimental ECD and calculated ECD of compounds **13-16**. **d** Corresponding deprotonated molecular ion peaks ([M-H]⁻ *m/z*) of compounds **13-16**: **13** ([M-H]⁻ m/z ~599.21094), **14** ([M-H]⁻ m/z ~599.21082), **15** ([M-H]⁻ m/z ~599.21082) and **16** ([M-H]⁻ m/z ~599.21075).

**Table 1 Cytotoxic activity of compounds 1-33 from SWFs against HUVECs.**

| Compounds | IC$_{50}$ (μM) | Compounds | IC$_{50}$ (μM) |
|---|---|---|---|
| 1 | 119.65 | 18 | 44.50 |
| 2 | 117.26 | 19 | 119.58 |
| 3 | 33.33 | 20 | 34.58 |
| 4 | 32.14 | 21 | 33.82 |
| 5 | 72.78 | 22 | 114.07 |
| 6 | 31.21 | 23 | 37.55 |
| 7 | 21.33 | 24 | 1.97 ± 0.31 |
| 8 | 40.77 | 25 | 109.18 |
| 9 | 23.64 | 26 | 52.81 |
| 10 | 43.21 | 27 | 13.04 ± 2.81 |
| 11 | 23.53 | 28 | 66.52 |
| 12 | 20.9 | 29 | 14.91 ± 1.38 |
| 13 | 31.81 | 30 | 9.38 ± 1.23 |
| 14 | 34.09 | 31 | 28.88 |
| 15 | 44.06 | 32 | 14.43 ± 1.66 |
| 16 | 36.61 | 33 | 15.04 ± 2.56 |
| 17 | 102.65 | Adriamycin | 2.54 ± 0.49 |

Results as the mean IC$_{50}$ in triplicate.

**Toxicity analysis of silexcrin B (2) and DPL (24) in vitro.** Regarding glycosylation associated with detoxification effects, we assumed that PIFs may be toxic to silkworm, but the bio-transformed GPIFs would have attenuated the toxicity. First, we screened the cytotoxicity of compounds **1**-**33** isolated from SWFs on the Normal Human Umbilical Vein Epithelial Cells (HUVECs) (Table 1). These PIFs from CTLs generally exhibited significant or moderate cytotoxic activity on HUVECs. Notably, DPL (**24**) showed the most significant cytotoxicity to HUVECs at 1.97 ± 0.31 μM, which was stronger than that of the control drug azithromycin, while the toxicity of the corresponding five GPIFs **1**-**5** was rather weak or even disappeared.

We also observed the growth and survival of HUVECs in two groups treated with silexcrin B (**2**) and DPL (**24**) (Fig. 5a). The growth state of HUVECs treated with silexcrin B (**2**) was still with normal cell morphology up to 80 μM, while the cells were obviously and dose-dependently damaged in the DPL (**24**)-treated group at less than 5 μM. Further study found that DPL (**24**) affected cell apoptosis and cycle arrest when HUVECs were treated with **24**, as determined by flow cytometry (Fig. 5b–e). This result implied that DPL (**24**) dose-dependently promoted HUVEC apoptosis and detained HUVECs in G1 phase, which may be interpreted by the toxic effect on DPL (**24**).

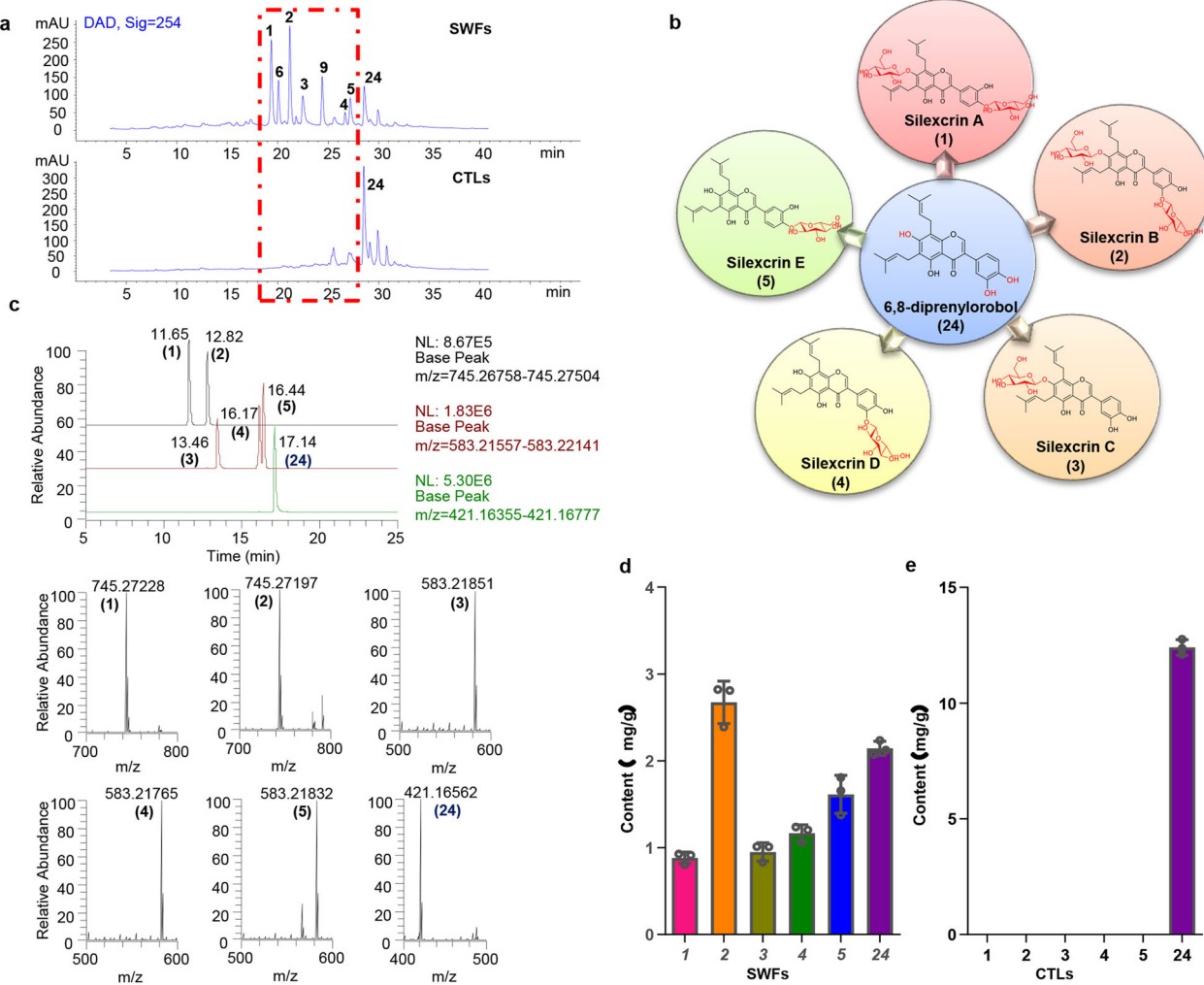

**Fig. 3 Comparative chemical profile analysis of SWFs and CTLs. a** HPLC qualitative analysis of the crude extracts of SWFs and CTLs. The difference peaks in SWFs corresponded to compounds **1-6** and **9**, which were the major distinct ingredients from CTLs, highlighted by the red outline. **b** The structure relationship of glycosylation among compounds **1-5** and **24**. **c** The retention times of silexcrins A-E (**1-5**) and DPL (**24**) were determined to be 11.65, 12.82, 13.46, 16.17, 16.44, and 17.14 min by LC–MS, respectively, and the deprotonated molecular ion peaks were **1** ([M-H]⁻ m/z ~745.27228), **2** ([M-H]⁻ m/z ~745.27197), **3** ([M-H]⁻ m/z ~583.21851), **4** ([M-H]⁻ m/z ~583.21765), **5** ([M-H]⁻ m/z ~583.21832), and **24** ([M-H]⁻ m/z ~421.16562). **d**, **e** Quantitative analysis results of **1-5** and **24** in SWFs and CTLs by LC–MS. Errors bars from **d**,**e** represent mean ± SD.

**Toxicity analysis of silexcrin B (2) and DPL (24) on *Galleria mellonella* in vivo.** Subsequently, we tested the toxicity of silexcrin B (**2**) and DPL (**24**) on *G. mellonella* (Fig. 5f). Silkworm and *G. mellonella* are both insects of Lepidoptera; thus, *G. mellonella* as a model insect[37],[38] were treated with 40 or 80 μg/d compounds and continuously injected for 7 days. *G. mellonella* in the DPL (**24**)-treated group at 80 μg/d started dying on the second day, and 50% survival rate was observed on the seventh day; in the silexcrin B (**2**)-treated group at the same drug dose, insects began dying on the fifth day, with a survival rate of 75% on the seventh day. A similar result occurred when the dose was 40 μg/d. Under the same dose and treatment duration, the DPL (**24**)-treated group had a lower survival rate than the silexcrin B (**2**)-treated group, and the survival rate decreased with increasing dose. The toxicity test on *G. mellonella* implied that the PIFs in CTLs had relatively high toxicity to silkworms, and these compounds were converted to glycosylated derivatives with greatly attenuated toxicity by the intestinal microbiota of silkworms. Thus, glucosylation, as an important detoxification pathway, played an important role in silkworm gut metabolism when silkworms were fed CTLs.

**B. subtilis as a probiotic influenced the silkworm gut flora.** As the composition of the gut microbiota was affected by diet change[14],[39], we supposed that *B. subtilis* supplementation in silkworm diets would have some beneficial effects on silkworm and gut microbes. Because *B. subtilis* with a clear genetic background was used as a common engineered bacterium and probiotic supplement[40–42], a probiotic trial using *B. subtilis* probiotic powder on two different breeds of silkworm was conducted, including control groups fed CTLs without any application of probiotic powder, and experimental groups fed CTLs with probiotic powder.

The intestinal contents dissected from living silkworms were sequenced by the 16S rDNA amplicon sequence method[43] to compare the changes in the gut microbial composition, relative abundance, and diversity among different groups with or without *B. subtilis*. First, the composition of the silkworm gut microflora at the genus level and even at the phylum level was changed, which was influenced by the addition of extraneous *B. subtilis* (Fig. 6a, b). Furthermore, it also increased the relative abundance of the genus *Bacillus* in the silkworm gut when adding *B. subtilis*, which increased from 1.19% to 2.12% in YSY-silkworm, increased

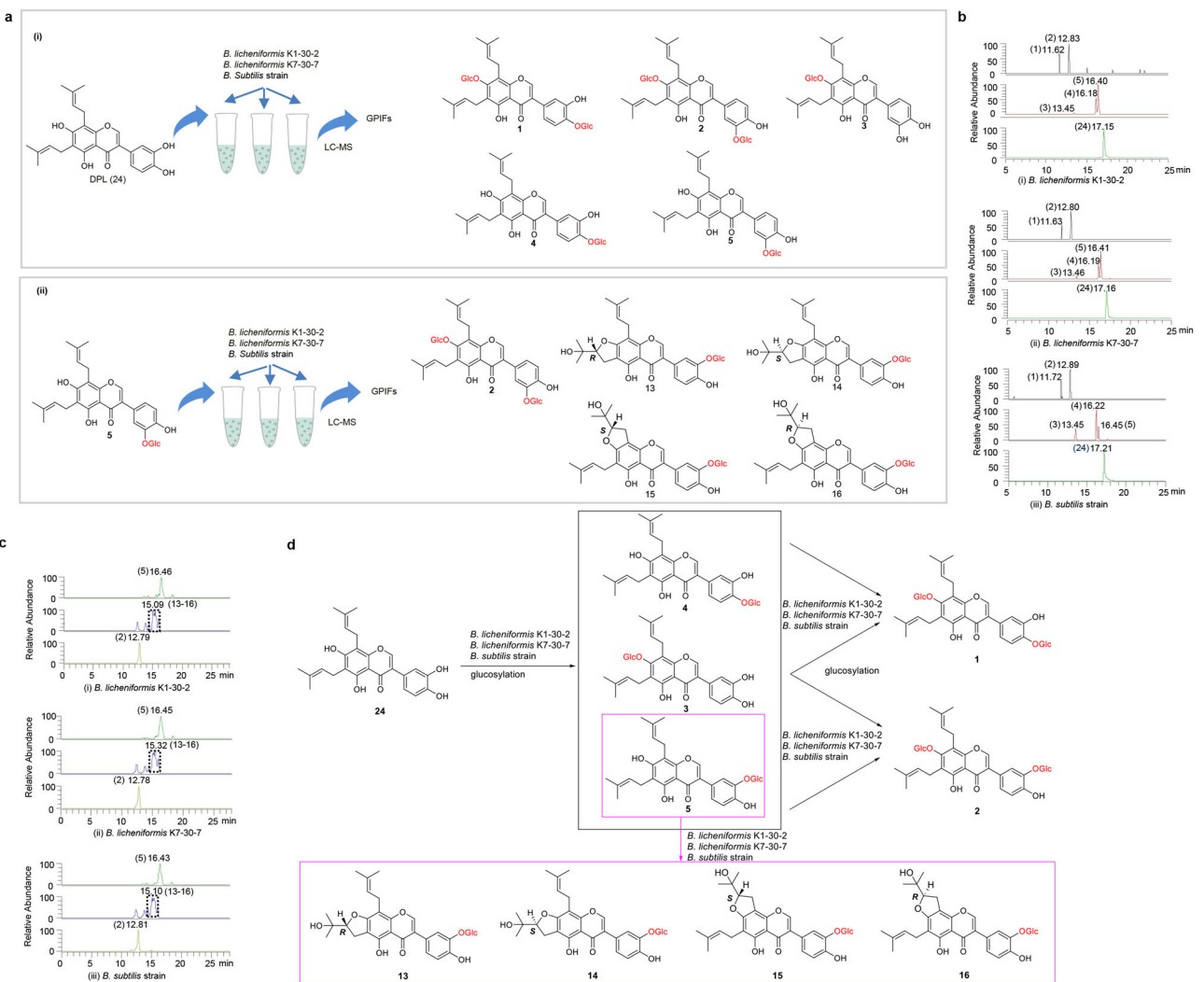

**Fig. 4 Microbial biotransformation of glycosylation of 5 and 24 by *Bacillus* species in vitro. a** Scheme of microbial biotransformation of (ii) **5** and (i) **24** by *Bacillus* species in vitro. Silexcrin E (**5**) and DPL (**24**) were used as substrates to feed with three *Bacillus* species, *B. licheniformis* K1-30-2, *B. licheniformis* K7-30-7 and *B. subtilis* strain. The converted products were detected by LC–MS. **b** Converted products from three *Bacillus* species with DPL (**24**) as substrate through microbial biotransformation in vitro detected by LC–MS: (i) Compounds **1-5** produced by *B. licheniformis* K1-30-2 were detected at 11.62, 12.83, 13.45, 16.18, 16.40 min, respectively. (ii) Compounds **1-5** produced by *B. licheniformis* K7-30-7 were detected at 11.63, 12.80, 13.46, 16.19 and 16.41 min, respectively. (iii) Compounds **2-5** produced by *B. subtilis* strain were detected at 11.72, 12.89, 13.45, 16.22 and 16.45 min, respectively. **c** Converted products from three *Bacillus* species with silexcrin E (**5**) as substrate through microbial transformation in vitro detected by LC–MS: (i) Compounds **2** and **13-16** produced by *B. licheniformis* K1-30-2 were detected at 12.79 and 15.09–15.35 min, respectively; (ii) the aforementioned products converted by *B. licheniformis* K7-30-7 were detected at 12.78 and 15.09–15.35 min, respectively; and (iii) produced by *B. subtilis* strain was detected at 12.81 and 15.09–15.35 min, respectively. **d** Proposed microbial biotransformation of GPIFs from SWFs by *Bacillus species* in the silkworm gut.

from 0.72% to 1.26% in *HK2*-silkworm (Fig. 6c). According to the alpha diversity analysis, the three indicators of Shannon_2, richness, and Chao1 illustrated that the species richness in the silkworm gut of the experimental groups with *B. subtilis* was higher than that in the control groups without *B. subtilis* (Fig. 6d). For example, the value of Shannon_2 increased from 3.12 to 3.98 in *YSY*-silkworm, and from 1.42 to 2.44 in *HK2*-silkworm. In addition, the Simpson, dominance, and equitability indicators also indicated that the species evenness in the silkworm gut was more even after adding *B. subtilis* (Fig. 6e). For example, the value of Simpson decreased from 0.40 to 0.21 in *YSY*-silkworm, and from 0.72 to 0.53 in *HK2*-silkworm. Therefore, it indicated that *B. subtilis* could colonise in the silkworm gut, changed the gut microflora composition, and increased the gut microbial diversity and overall evenness. But we did not explore a control group of other ingredients in probiotic powder without *B. subtilis*, nor

could we rule out the influence of other ingredients in the current study.

***B. subtilis* influenced the growth and development of silkworm and gut metabolites**. Additionally, silkworms in the probiotic trial were observed for growth and development in different groups, and the weight data of silkworms in different periods were recorded. As a result, we found a better growth state of silkworms in the experimental groups, and the average weight of silkworms fed *B. subtilis* in different periods were heavier than those not fed *B. subtilis* (Fig. 7a), and the silkworms fed *B. subtilis* were larger than those not fed *B. subtilis* (Fig. 7b). For the breed of *YSY*-silkworms, it shown that silkworms with *B. subtilis* supplementation were significantly heavier than those without *B. subtilis* supplementation (Fig. 7c). Hence, it was, to a certain

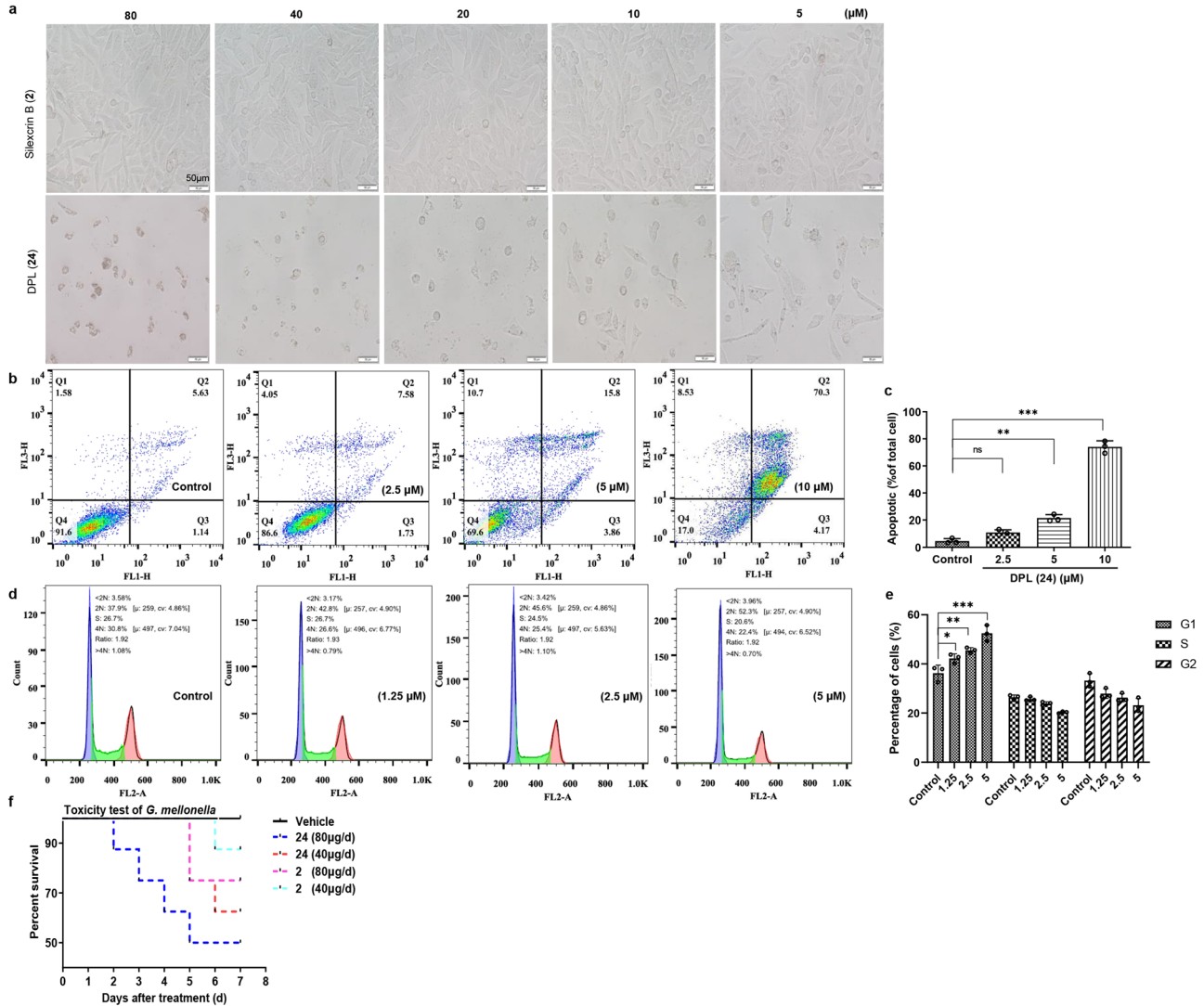

**Fig. 5 DPL (24)- or silexcrin B (2)-induced cytotoxicity in HUVECs in vitro and *G. mellonella* in vivo. a** Growth state of HUVECs induced by different concentrations of silexcrin B (**2**) or DPL (**24**) in vitro. **b** The HUVEC line was treated with 0, 2.5, 5, or 10 μM DPL (**24**), and cell apoptosis was detected by flow cytometry analysis coupled with Annexin V-FITC/PI staining. **c** Quantification panel shows the statistical analysis of cell apoptosis in a dose-dependent manner. (one-way ANOVA, $n = 3$, $P = 0.0687 > 0.05$, $**P = 0.003 < 0.01$, $***P < 0.001$ vs. the control group. The error bars are reported as mean ± SD.) **d** Cells were treated with 0, 1.25, 2.5, or 5 μM DPL (**24**), and cell cycle arrest was also detected by flow cytometry analysis coupled with PI staining. **e** Quantification panel shows the statistical analysis of cell cycle arrest. (one-way ANOVA, $n = 3$, $*P = 0.0496 < 0.05$, $**P = 0.0053 < 0.01$, $***P < 0.001$ vs. the control group. The error bars are reported as mean ± SD). **f** Toxicity test of silexcrin (**2**) and DPL (**24**) on *G. mellonella in* the survivorship curve.

extent, indicated that feeding *B. subtilis* to silkworms can promote their growth and development, notably increasing the weight of silkworms.

Then, quantitative analysis of principal components **1-5** and **24** by HPLC was performed (Fig. 7d, e). According to the quantitative results of HPLC, the ratio of each peak area of **1-5** and **24** to the total peak area of the six components was used as an indicator to measure the changes in the relative contents of the six components in SWFs to illustrate the influence of exogenous *B. subtilis* on the changes in gut metabolites from silkworm. As shown in Fig. 7e, B. *subtilis* affected the contents of gut metabolites in silkworms. After adding *B. subtilis*, the total relative contents of five principal GPIFs **1-5** in SWFs increased in *YSY*-silkworms (Fig. 7f), which was 2.0% greater than that in silkworms not fed *B. subtilis*, while the relative content of the aglycone DPL (**24**) decreased. This observation may be associated with the increase in the relative abundance of *Bacillus* in the *YSY*-

silkworm gut. However, the total relative content of **1-5** in SWFs from *HK2*-silkworms increased slightly by 0.2% compared with that in silkworms no fed *B. subtilis*. This result suggested that *YSY*-silkworms may be more susceptible to *B. subtilis* than *HK2*-silkworms.

The phenomenon that *B. subtilis* supplementation in silkworm enabled an increase in the total content of GPIFs and a decrease in the corresponding toxic aglycone content in SWFs, which may be also relevant to detoxifying the toxic PIFs in CTLs to protect hosts from adverse ingredients.

## Discussion

We discovered a series of GPIFs abundant in SWFs, which are oxy-glycosylated products at one or two sites of the 7-, 3′-, or 4′-OH of the prenylated isoflavone skeleton. Given the few naturally occurring GPIFs in nature[44], SWFs from silkworms fed

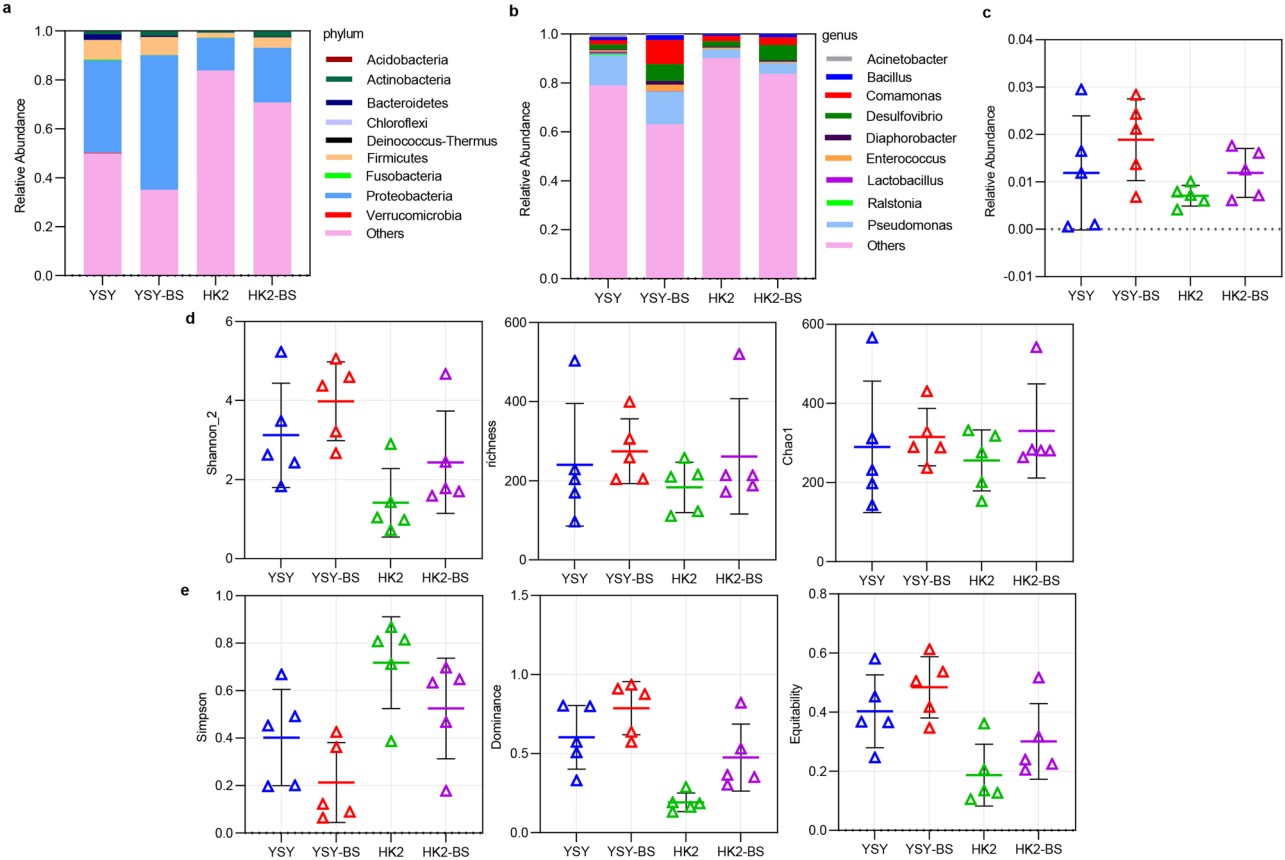

**Fig. 6 16S rDNA bacterial sequencing analysis in the silkworm midgut.** Relative abundances of silkworm gut bacteria at the phylum (**a**) and genus (**b**) levels in the four different groups. There were 5 silkworms in each group in four different groups. (YSY refers to the group of *YSY*-silkworms; HK2 refers to the group of *HK2*-silkworm; BS means adding additional *B. subtilis* to CTLs to feed silkworm.) **b** Relative abundance of *Bacillus* species in the four groups of silkworm midguts. **c** Boxplots of Shannon_2, richness and Chao1 were the indicators of species richness on behalf of the higher the value, the higher the species richness, and the value of the median line in boxplots illustrating the species richness of experimental groups with *B. subtilis* higher than control groups without *B. subtilis*; **d** Simpson, dominance and equitability were the indicators of species evenness. Simpson is representative of the smaller the value, the higher the species evenness, and the value of the median line illustrating the species evenness of experimental groups with *B. subtilis* was more even than those groups without *B. subtilis*. For dominance and equitability, the higher the value is, the more evenly the species evenness is. The data from **c**–**e** are reported as the mean ± SD.

on CTLs consequently become an abundant natural source for these compounds. Moreover, we found that the major differences in ingredients between SWFs and CTLs were GPIFs. Subsequently, preliminary evidence by HPLC–MS quantitative analysis proved that these GPIFs were not present in CTLs themselves but produced by silkworm gut metabolism. An in vitro microbial biotransformation of DPL (**24**) by three intestinal strains, *B. subtilis*, *S. sciuri* and *E. hormaechei* isolated from silkworm gut further suggested that the formation of abundant GPIFs in SWFs was closely related to silkworm microbes.

It has been reported that some microbes, such as *Bacillus* species with GTs, have excellent capacity for transglycosylation with broad substrate specificity, especially prenylated phenolic components with higher affinity[44,45]. In recent years, some GTs from *Bacillus* species have been widely used to glycosylate flavonoids[46]. Drawing wide attention to *O*-glycosylated flavonoids with relatively high proportions in natural total flavonoid glycosides, extensive studies discovered that GTs from *B. subtilis* ATCC 6633 (BsGT110)[47], *B. subtilis* 168 (Bs-YjiC)[48], *B. licheniformis* (Bl-YjiC)[49,50] and *Bacillus cereus* (MgtB, BcGT1, and BcGT-3)[51–53] were able to catalyse the glycosylation reaction, which attached glycosyl moieties to hydroxyl groups of flavonoids to biosynthesize flavonoid-*O*-glucosides. To exemplify this, we proved that *Bacillus* species could convert DPL (**24**) into the

GPIFs silexcrins A-E (**1-5**). In addition, the formation of the epimers silexcrins M-P (**13-16**) having mutated aglycone skeletons was also confirmed by microbial biotransformation.

Glycosylation not only can enhance the structural stability and hydrophilicity of chemical compounds but is also considered a direct or indirect detoxification process[54–57]. In our present work, the toxicity tests of DPL (**24**) on *G. mellonella* proved that PIFs in CTLs had toxic effects on herbivores, which may affect the growth and development of silkworms. Further testing of DPL (**24**) with HUVECs showed that it induced cell apoptosis, blocked cells to stay in the G1 phase, and caused severe cell damage. This study revealed that glycosylation was an important detoxification pathway for converting toxic PIFs in CTLs into GPIFs with attenuated or no toxicity. According to some studies reported, silkworms fed on CTLs make difference in growth and development, cocoons and the quality of the silk (Supplementary Fig. 217) compared with those on mulberry leaves, which may be influenced by the toxic components from CTLs[15,17,18]. Therefore, PIFs, a subclass of characteristic phytochemicals abundant in CTLs, were confirmed to be adverse ingredients harmful to silkworm growth, which explains why silkworms have relatively low adaptability when fed CTLs.

Recent studies also proposed a potential strategy of a probiotics trial, which revealed that the intestinal microbes could change

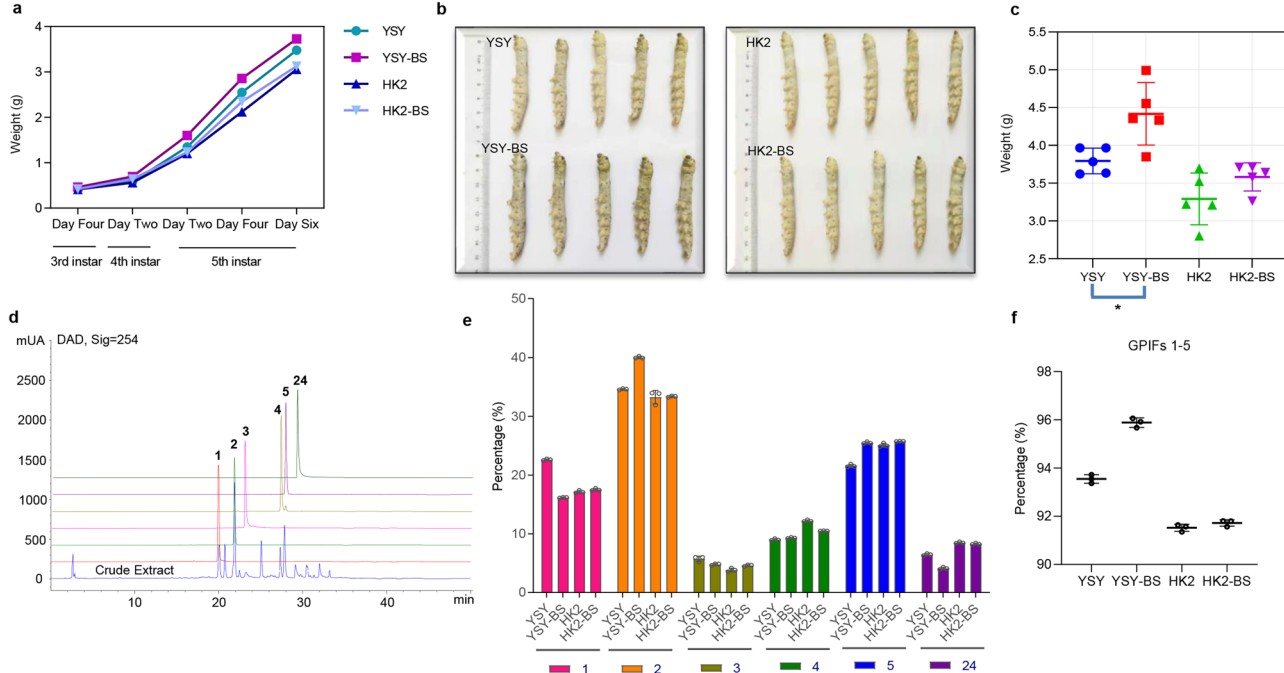

**Fig. 7 Evaluation of the influence on the growth and development of silkworm and intestinal metabolites in probiotic trials. a** Average weight changes of all silkworms involved in the probiotic trial in four different groups at different growth stages. **b** Comparison of the sizes of silkworm bodies picked from 5 of each group at random in the probiotic trial. **c** Comparison of the weight of silkworms picked 5 at random in four different groups (Student's t test, two-tailed, $n = 5$, *$p = 0.0143 < 0.05$). **d** HPLC quantitative analysis of GPIFs **1-5** and aglycone **24** detected at $\lambda = 254$ nm. **e** Quantitative analysis results of the relative content of **1-5**, **24** from SWFs in four groups by HPLC in bar graph. The vertical coordinate refers to the ratio of the content of each compound (**1-5** and **24**) accounting for the total content of these six compounds, which reflects the relative content of ingredients in SWFs. **f** Comparison of the relative contents of GPIFs **1-5** from SWFs in the four different groups. The data from **c**, **e**, **f** are reported as the mean ± SD.

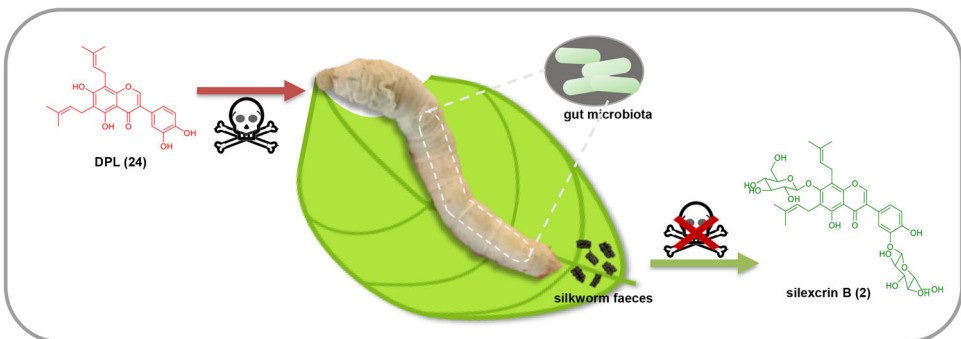

**Fig. 8 Presumptive detoxification mechanism by glucosylation of DPL (24) from CTLs under the influence of silkworm gut microbiota.** When silkworm feeds on CTLs, the toxic component DPL (**24**) in CTLs would enter the silkworm gut and would be converted into GPIFs, like silexcrin B (**2**) with greatly attenuated toxicity under the influence of silkworm gut microbiota.

accordingly when *Corynebacterium* and *Bacillus* species were used as probiotics to verify the microbial interactions in the treefrog gut[58]. Here, we found that *B. subtilis*, as a probiotic added to the silkworm gut, played important roles in changing the intestinal microbiota composition, increasing the richness and evenness of intestinal bacteria and manifesting an unexposed metabolic detoxification strategy through phytochemical glycosylation in silkworms. Importantly, a beneficial promotion influence on the growth and development of silkworm was also observed. Given that the probiotic *B. subtilis* was used to promote the growth and development of animals and was even shown to be beneficial for the prevention and treatment of human diseases[34,59–62], probiotic *Bacillus* species are also expected to be developed into intestinal probiotic preparations for silkworms.

However, the limitations in the current study are that no more endogenous gut microbiota other than three isolates, *B. subtilis*, *S. sciuri* and *E. hormaechei* from silkworm gut were investigated here. As is reported that the differences in diet could affect the composition of insect gut microbiota[14,39], we could not clear if other ingredients in probiotic powder also might influence silkworm gut microbiome composition.

Here, it is worth noting that we revealed that the silkworm gut microbiota evolved the ability to detoxify food ingredients by converting them into glycosylated derivatives (Fig. 8). It lays a basis for future related research to improve the adaptability of silkworms fed on CTLs and provides a potential direction for the further development of probiotics applied in microbial preparations suitable for silkworms.

## Methods

**Experimental materials, chemicals and reagents**. CTLs and SWFs from silkworms fed CTLs were collected from Linyi, Shandong, China. A probiotic dry powder preparation of *B. subtilis* was purchased from Shandong Yihao Biotechnology Co., Ltd. (China). The reagents for HPLC or LC-MS analysis were of chromatographic grade, and other reagents were of analytical grade.

**Extraction, isolation and identification of compounds**. Air-dried SWFs (1.8 kg) were extracted by refluxing them with 95% ethyl alcohol three times for 3 h each time. The evaporated crude extract suspended in $H_2O$ was extracted by petroleum ether, ethyl acetate and n-butyl alcohol successively. Briefly, the ethyl acetate extract was sequentially partitioned with a gradient of MeOH-$H_2O$ (3:7 to 1:0), $CH_2Cl_2$-MeOH (200:1 to 0:1), MeOH (100%) and a MeOH-$H_2O$ elution system via an MCI gel column, a silica gel column (100−200 mesh), a Sephadex LH-20 chromatography and a semipreparative HPLC (SHIMADZU LC-20AT, DAD-detector, Shim-pack GIS-C18 (5 μm, 10×250 mm)). Thirty-three GPIFs were purified, and the elucidation of silexcrins A-U (**1-21**) as well as NMR data, HRESIMS, ECD, UV, and IR spectra are shown in the supplementary materials.

**General experimental procedures**. NMR spectra were acquired by Avance DRX-400 and 600 spectrometers (Bruker, Germany, Dimethyl Sulfoxide-d$_6$ with 0.3% internal standard of TMS as solvent). The UV spectra were obtained through the UV-2550 spectrophotometer (Shimadzu, Japan) and IR spectra were gained by Nicolet iN10 Microinfrared spectrometer (Thermo Fisher Scientific, America). Chirascan Circular dichroism spectrometer (Applied Photophysics, United Kingdom) were used to obtain the compounds of ECD spectra. Optical rotations were measured by Modular Circular Polarimeter MCP 200 (Anton Parr, Austria, MeOH as solvent, 20 °C). High-resolution mass spectra were obtained by Thermo Fisher Q-Exactive Orbitrap Mass Spectrometer (Thermo Fisher Scientific, America).

**Quantitative analysis by LC–MS**. The quantitative analysis of the changes in the contents of characteristic chemical constituents silexcrins A-E (**1-5**) and DPL (**24**) from SWFs and CTLs were separately carried out by LC-MS with the following analytical conditions: instrument, Thermo Fisher Q-Exactive Orbitrap, UtiMate 3000, Dim. (100×2.1 mm); detector, DAD-3000; ionisation source, ESI; flow rate, 0.3 mL/min; and injection volume, 2.0 μL. Gradient elution was performed for the MeOH-$H_2O$ elution system: 25:75 from 0.0–1.0 min, 25:75 to 95:5 from 1.0-20.0 min, 95:5 from 20.0-24.0 min, 95:5 to 25:75 from 24.0-25.0 min, and 25:75 from 25.0-28.0 min. The six silexcrins A-E (**1-5**) and DPL (**24**) were mixed together to prepare a standard solution of each compound at a concentration of 1 μg/mL. The mixed standard solution was serially diluted to the concentrations of 1000, 500, 250, 100, 50, 25, 10, 5 and 2.5 ng/mL for each compound in each mixture solution. These nine standard mixture solutions were applied to prepare the standard curves of the six components (Supplementary Fig. 218). The methanol ultrasonic extracts of SWFs and CTLs were separately prepared to a concentration of 200 μg/mL by LC-MS under the above conditions. Each group was analysed in triplicate. Data obtained from the analysis were used to detect the contents of six compounds in SWFs and CTLs by Thermo Scientific Xcalibur software.

**Qualitative and quantitative analysis by HPLC**. The qualitative analysis of different constituents from the crude extracts of SWFs and CTLs was performed by HPLC under the same chromatographic conditions (Agilent 1260, DAD-detector, Eclipse XDB-C18 column (5 μm, 4.6 × 250 mm), gradient elution from the MeOH-$H_2O$ elution system, 30:70 to 100:0 in 30 min). The quantitative analysis of the content changes in silexcrins A-E (**1-5**) and DPL (**24**) in fresh SWFs were separately carried out by HPLC with the above chromatographic conditions. There were four groups of SWFs (YSY, YSY-BS, HK2, HK2-BS) produced by two breeds of silkworm with or without *B. subtilis*, all of which were prepared to 50 mg/mL. Moreover, silexcrins A-E (**1-5**) and DPL (**24**) dissolved in chromatographic-grade methanol were qualitatively analysed to confirm the retention time under the above chromatographic conditions. The six chromatographic peak areas attained were processed to compare the relative contents of the six compounds in the four different excrements. All fresh SWFs after lyophilization in addition to water were ground into powder.

**ECD calculations details of 13-16**. Theory and Calculation Details[63]: The calculations were performed by the Gaussian 09 program package. The potential energy surface scanned were used by semi-empirical AM1 method and a DFT approach B3LYP/6-31 G (d), and the geometries of all ground-state conformations were further optimised at 298.15 K. Then, these minima and calculations of room-temperature free energies were confirmed by calculations of their harmonic frequency analysis. The electronic excitation energies and rotational strengths in the gas phase for the first 60 states were calculated by Time-dependent density functional theory. After summed the rotatory strengths and energetically weighted based on Boltzmann statistics, the final ECD spectra were obtained by the following Equation 1:

$$\Delta\varepsilon(E) = \frac{1}{2.296 \times 10^{-39}} \frac{1}{\sigma\sqrt{\pi}} \times \sum_i \Delta E_i R_i e^{-[(E-\Delta Ei)/\sigma]^2}$$

$\sigma$ is the width of the band at $1/e$ height ($\sigma = 0.1$ eV). $\Delta E_i$ is the excitation energies. $R_i$ is rotatory strengths for transition.

**Microbial biotransformation in vitro**. *B. licheniformis* K1-30-2 and *B. licheniformis* K7-30-7, used for microbial biotransformation in vitro were selected previously in our laboratory. *B. subtilis* strain was isolated from silkworm in our work. DPL (**24**) and silexcrin E (**5**) were used as substrates to feed on three *Bacillus* species. The experimental groups with substrate and control groups without substrate were set up, and each group was tested in triplicate. Three *Bacillus* species were activated by culture on agar plates, and activated isolates were subcultured into Luria-Bertani (LB) solution with 5 mL in Eppendorf tubes as culture medium to provide the source of glucose on a 37 °C constant temperature shaking table for 24 h. After the bacterial solution had been incubated, DPL (**24**) or silexcrin E (**5**) (1 mg dissolved in 0.5% DMSO) as substrate was added to the experimental groups and continuously coincubated for another 24 h at 37 °C. Then, the bacteria in solution were crushed by ultrasound and extracted by ethyl acetate three times. The lysates were analysed by LC-MS (under the same conditions as mentioned above) to detect the microbial biotransformation products of the three *Bacillus* species in vitro.

**Probiotic test**. Two of the silkworm breeds, *YeSanYuan* silkworm (*YSY*-silkworm) and *HuaKang 2* silkworm (*HK2*-silkworm), chosen in our study were provided by the Runfa Institute of *Cudrania tricuspidata*, Linyi (Shandong, China). The control groups (YSY, HK2) in probiotic test were fed CTLs without any application of *B. subtilis* probiotic powder, and the experimental groups (YSY-BS, HK2-BS) were fed CTLs with *B. subtilis* probiotic powder. There was a total of 400 silkworms in the experiment, 100 in each group. They were fed on CTLs beginning at the hatched silkworm stage. From the third instar on, silkworms in the experimental groups were fed CTLs supplemented with *B. subtilis* preparations by spraying. The dry powder preparation of *B. subtilis* was diluted in water to 1:1000 g/mL, which was added moderately to CTLs fed to silkworms in experimental groups three times a day. Simultaneously, the changes in silkworm body weight were recorded. Due to the highest bacterial population recorded in the silkworm digestive tract when fifth-instar[64], five living silkworms in each group of two breeds were randomly selected. After surface sterilisation of silkworm bodies with 75% ethyl alcohol, they were dissected at the silkworm midgut. The midgut contents were taken by sterile syringes and stored at −80 °C.

**Bacterial community diversity by 16S rDNA amplicon sequence analysis**. The samples of silkworm midgut contents were derived from *YSY*-silkworm and *HK2*-silkworm, four groups named YSY, YSY-BS, HK2, HK2-BS, every group with five parallel individuals, and a total of 20 samples. The following steps were conducted by Luojie (Jinan) Biological Medicine Co., Ltd. (Shandong, China). Nucleic acid in the extracted samples was analysed using DNeasy PowerSoil Pro Kit (Qiagen, Cat No. 47016). After the integrity and concentration of nucleic acids were determined, DNA samples were amplified using high-fidelity enzymes, and an Invitrogen Qubit 4.0 fluorimeter was used for concentration quantification. The library was constructed using the KAPA Hyper Prep Kit and sequenced using Illumina NovaSeq to obtain raw data. Zero-radius operational taxonomic units (ZOTUs) were constructed by effective tags obtained after a series of data separation, primer removal, PE read splicing, tags with quality and length, filtering and interception, and chimaerism removal. Bacterial alpha-diversity and beta-diversity analyses were obtained from ZOTU. 16 S rDNA amplicon sequencing[65] was used to analyse the difference in intestinal microbes between the experimental groups and control groups as well as the changes in the composition of intestinal microbes in our experiment.

The genome DNA extraction: Total genome DNA from silkworm intestinal bacteria was extracted by DNeasy PowerSoil Pro Kit (Qiagen, Cat No. 47016) according to the extraction process as follows: Lysis buffer is added to the sample in mixed zirconium bead tubes and bead beating is performed by a benchtop vortex with bead tube adapter to homogenise. The crude lysate is subjected to inhibitor removal for cleanup and the purified lysate is mixed with an equal volume of DNA binding solution. The mixed system passed through a silica spin filter membrane which is washed with a two-step washing regime. A 10 mM Tris elution buffer is then used to elute the Silica-bound DNA.

PCR reaction system and cycle procedures: 16S rRNA genes of distinct regions (16S V3-V4) were amplified used specific primer (341 F (5′-CCTAYGGGRBGCA SCAG-3′), and 806 R (5′-GGACTACNNGGGTATCTAAT-3′) with the barcode. The PCR reactions were carried out with 15 μL of Phusion® High-Fidelity PCR Master Mix (New England Biolabs); 0.2 μM of forward and reverse primers, and about 10 ng template DNA. Thermal cycling consisted of initial denaturation at 98 °C for 1 min, followed by 30 cycles of denaturation at 98 °C for 10 s, annealing at 50 °C for 30 s, and elongation at 72 °C for 30 s. Finally, 72 °C for 5 min.

**Isolation and identification of gut bacteria from silkworm gut**. We isolated the silkworm intestinal bacteria according to the reported method[36] with modification. Briefly, five silkworms starving for 24 h were selected. After sterilising with 75% alcohol on the silkworm body surface, we dissected silkworms and obtained the intestinal contents of silkworms. It was added into 10 mL EP tube containing 5 mL

sterile water under aseptic operation. The intestinal fluid was diluted to $10^{-1}$, $10^{-2}$ and $10^{-3}$ by a 10-fold dilution method. The intestinal fluid at each concentration of 100 µL was uniformly coated on NA medium plate and cultured at 37 °C for 12 h. Colonies with different morphology were selected for further striation and purification until they were detected as a single strain by microscopy and sent to Sangon Biotech (Shanghai) Co., Ltd. for 16S rDNA sequence and analysis.

The genome DNA extraction: The silkworm strains genomic DNA were extracted by Ezup Column Bacteria Genomic DNA Purification Kit (Sangon Biotech, Cat No. SK8255) according to the extraction process as follows: The overnight cultured bacterial solution (1 mL) was added to 1.5 mL centrifuge tube, centrifuged at room temperature at 8000 rmp to discard supernatant and collect bacteria. The gram-positive bacteria solution is added with 180 µL lysozyme solution (20 mg/mL), resuspended, and bathed at 37 °C for 30-60 min. The gram-negative bacteria solution is added with 180 µL Buffer Digestion. Proteinase K solution (20 µL) is added to the bacteria collected, mix well and water bath at 56 °C for 1 h until the cells were completely lysed. Subsequently, Buffer BD (200 µL) and anhydrous ethanol (200 µL) are added in sequence to the lysate and thoroughly mixed. The mixture is loaded to the adsorption column for 2 min, then centrifuged at room temperature at 12,000 rpm for 1 min, and the waste liquid in the collection tube was emptied. Add PW Solution (500 µL) and centrifuge at 10,000 rpm for 30 s to drain the filtrate. Next, add Wash Solution (500 µL), centrifuge and remove the filtrate. The adsorption column was placed into a 1.5 mL centrifuge tube, added with CE Buffer (50–100 µL) and left for 3 min, centrifuged at 12,000 rpm for 2 min at room temperature, and DNA solution was collected.

PCR reaction system and cycle procedures: The extracted bacterial genomic DNA was used as templates. PCR amplification was performed using universal primers 27 F (5′-AGA GTT TGA TCC TGG CTC AG-3′) and 1492 R (5′-GGC TAC CTT GTT ACG ACT T-3′) of bacterial 16 S rRNA gene. 0.5 µL genomic DNA template (20-50 ng/µL), 0.5 µL forward and reverse primers (10 µM), 5 µL 10×PCR buffer (Mg$^{2+}$ plus), 4 µL dNTP Mixture (2.5 mM L$^{-1}$) and TaqDNA polymerase (5 U µL$^{-1}$) 0.2 µL, filled with sterilised ultra-pure water to 25 µL. Predenaturation at 94 °C for 4 min, denaturation at 94 °C for 45 s, annealing at 55 °C for 45 s, extension at 72 °C for 1 min, 30 cycles, and the repair extension at 72 °C for 10 min; then store at 4 °C. The amplified products were purified by 1% AGAR gel electrophoresis (BBI, Cat No. AB0014), and recovered by SanPrep column DNAJ gel recovery kit (Sangon Biotech, Cat No. SK8131).

**Toxicity test on *G. mellonella*.** To exploit the influence of glycosylation of the prenylated isoflavones, a toxicity test on *G. mellonella* was implemented. *G. mellonella* as an insect model was used to test the toxicity of silexcrin B (**2**) and DPL (**24**) in vivo based on a described methodology[38] with modification. Briefly, a total of forty larvae of *G. mellonella*, weighing 0.28-0.35 g, were selected prior to the experiment and randomly divided into five groups, each group of 8, including vehicle control (with solvent served as controls), silexcrin B (**2**) (40 µg/d for each larva), silexcrin B (**2**) (80 µg/d for each larva), DPL (**24**) (40 µg/d for each larva), and DPL (**24**) (80 µg/d for each larva). Silexcrin B (**2**) and DPL (**24**) were dissolved in 10 µL solution with 5% DMSO, 45% PEG and 50% normal saline. Larvae in the control or four treated groups were injected via the last right pro-leg with 10 µL of the prepared drug solution and injected continuously for 7 days. Then, they were incubated at 35 °C in the dark. The survival of silkworms was monitored daily for 7 days. The survival curve was drawn by GraphPad Prism 7, and the log-rank (Mantel−Cox) survival analysis model was used.

**Cytotoxicity test by MTT assay.** HUVECs were obtained from the Shanghai Institute for Biological Sciences (SIBS), China Academy of Sciences (China). According to a previously reported method[66], the culture conditions of HUVECs were RPMI-1640 (HyClone) medium consisting of 10% FBS (Sijiqing Company Ltd.), 100 units/mL penicillin G and 100 µg/mL streptomycin in an environment with 5% CO$_2$ at 37 °C. Cells from the HUVEC line were first seeded into 96-well plates at $3-5 \times 10^3$ cells/well and allowed to adhere overnight. Then, different concentrations of 33 compounds and the control drug adriamycin were added to continue incubation for the indicated time. MTT (5.0 mg/mL) solution with 12 µL was added, followed by incubation for another 4 h at 37 °C. Then, MTT was removed from the medium with DMSO before 150 µL/well was added to the purple formazan crystals. The optical density was detected by a microplate reader (Bio-Rad 680) at 570 nm and calculated into IC$_{50}$ values using GraphPad Prism 7 to evaluate the cell viability. All experiments were performed in triplicate.

Through comparison to the vehicle control, the cell viability inhibitory ratio was calculated by the following Equation 2:

$$\text{Cell viability ratio}(\%) = \frac{A570\ sample - A570\ blank}{A570\ control - A570\ blank} \times 100\%$$

**Analysis of cell apoptosis and cell cycle arrest by flow cytometry.** Cell apoptosis and cell cycle arrest were analysed by flow cytometry according to the literature[66–68] with minor modifications as followed. Briefly, after overnight incubation, the cells were exposed to 5% serum medium with concentrations of 0, 2.5, 5, and 10 µM DPL (**24**) for 24 h and then harvested and washed with PBS. After the supernatant was removed by centrifugation, the cells were resuspended in 400 µL of binding buffer and incubated at room temperature in the dark for 15 min

with 5 µL of Annexin V-FITC, followed by 5 µL of PI (50 mg/L) for another 5 min. The apoptotic ratio was analysed by flow cytometry (Becton Dickinson, USA) using WinMDI 2.9 software. HUVECs were treated with 0, 1.25, 2.5, or 5 µM DPL (**24**) for 24 h, and cell cycle arrest was detected by flow cytometry analysis coupled with PI staining.

**Statistics and reproducibility.** Each experiment was performed with at least three biologically independent samples for statistics and reproducibility ($n \geq 3$). The results were used for analysis of variance, and differences between means were evaluated by Student's t test between two groups and one-way ANOVA among multiple groups. $P < 0.05$ was regarded as statistically significant. All the data are reported as the mean ± standard deviation (SD).

**Reporting summary.** Further information on research design is available in the Nature Portfolio Reporting Summary linked to this article.

## Data availability

All datasets supporting the findings of this study can be found in the figures, tables and supplementary information files. The numerical source data used to generate the graphs are available in Supplementary Data 1. The overall gut microbiome 16S rRNA-seq raw data are deposited in Sequence Read Archive with the accession code PRJNA985806. The raw sequence data of 16S rDNA sequences have been made publicly available at GenBank under the accession code OR125545 (*B. subtilis* strain), OR125546 (*S. sciuri*) and OR125547 (*E. hormaechei*). The structural elucidation of compounds 1-21 can be seen in Supplementary Notes 1 for details, and the physicochemical properties of compounds 1-21 can be seen Supplementary Note 2. All data are available from the corresponding author upon reasonable request.

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

## Acknowledgements

This work was supported by the National Natural Science Foundation of China (No. 82173703, 81874293), National Key R&D Program of China (No. 2019YFA0905700), Major Basic Research Program of Shandong Provincial Natural Science Foundation (No. ZR2019ZD26) and the Foundation for Innovative Research Groups of State Key Laboratory of Microbial Technology (No. WZCX2021-03). We are grateful to the staff at the Analytical and Testing Centre of Shandong University for collecting the spectro-scopic data.

## Author contributions

S.Y. and H.L. designed the study. S.Y., Y.S., J.S., J.Z. and M.S. carried out the experiments. S.Y., H.L., Y.S., W.C., J.Z., Y.Q., C.Z., Y.T. and M.Z. analysed the data; S.Y. and H.L. prepared the manuscript.

## Competing interests

The authors declare no competing interests.
