## [Peer Review File · Communications Biology]

Reviewers' comments:

Reviewer #1 (Remarks to the Author):

The authors studied the secondary metabolites in *Cudrania tricuspidata* leaves and in the faeces of silkworms that were feeding on them. Apparently the isoflavonoids of the leaves were found to be converted into glycosides in the frass. The authors assume that the glucosylation is a mean of detoxification.

The chemical experiments and structure elucidations are state of the art.

The biological experiments show some shortcomings which need to be addressed in a revision:

1. You should analyse whether the isoflavonoids are taken up by the insect larvae; check if you find them in the integument or body tissues. It could be that a detoxification also occurs inside the insect and not only my gut microbes.
2. To evaluate toxicity in cell cultures or insects we need IC50 values; I could not find precise values in the MS. Precise IC50 values are needed.
3. The authors should better consult the existing literature on the chemoeology of plant insect interactions. There are many more relevant mechanisms for an insect to deal with toxic secondary metabolites; discuss the function of ABC transporter, non-adsorption of toxins and rapid passage through the intestines or sequestration and target site inactivation.
4. The presentation and discussion of the microbiome data are inadequate
5. The English needs improvement by a native speaker.

Reviewer #2 (Remarks to the Author):

Dear Authors

The manuscript by Hongxiang Lou and co-authors investigates the capacity of the silkworm gut microbiome to biotransform prenylated isoflavones (PIFs) into less toxic glycosylated derivatives (GPIFs). CTLs are highly concentrated in *Cudrania tricuspidata* leaves, which is fed to the silkworm in order to strengthen the quality of produced silk. The authors analysed the content of isoflavones in the silkworm diet and faeces, and concluded that the gut microbiota, and in particular *Bacillus* sp. contribute to the isoflavones glycosylation, what provides a detoxifying effect to the host, and promotes its better growth and development, essentially leading to the improved silk production. Major part of the manuscript is devoted to the chemical analysis of the metabolites in the diet and insect faeces with techniques such as HPLC, LC-MS and NMR. This part of study is very well described, in terms of the methods used and generated results, and extensive supplementary data is also provided. I acknowledge the authors for this detailed analysis! The second part investigates the gut microbiome basically through the community profiling with the use of the 16S rRNA gene amplicon sequencing. In addition, some in vitro studies on the toxicity of PIFs on isolated *Bacillus* species and the capacity of these bacteria to biotransform PIFs, as well as and a bio-augmentation of gut microbiota with probiotic *Bacillus* sp. were also proposed. The study is quite interdisciplinary, however a strong focus is given to the chemical analysis, while my expertise in this domain is rather limited, therefore I will more focus on the microbiome-related aspects. I hope the other reviewers provide a constructive revision of this part of the study.

My major concern with regards to the microbiome analysis, is that there is actually a lack of evidence that the *Bacillus* species are the main microbes involved in PIFs biotransformation. I don't question their ability to do it, and the authors have showed it via the in vitro assays, and also by evidencing the beneficial effect on the host of gut microbiota bio-augmentation with *Bacillus* sp. However, in general, *Bacillus* sp. doesn't seem to dominate the gut microbiota of the silkworm (see Fig. 6). While other

phyla that are much more abundant in the gut were not considered here at all. Also, most if not all bacteria encode in their genomes multiple glycosyltransferases. Therefore, a sentence in lines 123-125 doesn't bring real evidence that these species are key to the biotransforming activity of the gut microbiome. In my opinion the authors a priori decided that *Bacillus* sp are important microorganisms in the silkworm microbiome, contributing to the detoxification effect, and designed the whole experimental part around this assumption. This is not a big issue, however I would clearly indicate it in the manuscript that the microbial analysis was targeted towards the analysis of the detoxifying activity of *Bacillus* sp., and not the gut microbiome. In addition, there is also no proof, that *Bacillus* species present in the silkworm gut are the same or at least closely related to the strains employed in the in vitro tests. I could not spot any 16S rRNA gene sequence comparison pointing to this. I think this study is of general interest and might be of high industrial relevance for silkworm breeders. Even though the scientific evidence on the establishment of the administrated probiotic *Bacillus* strain in the gut microbiome is relative weak, the observed health beneficial effect of the probiotic powder used can have important applied implications. To my opinion, any statistical analysis used is correctly employed, however some concern about the microbiome characterisation is given below.

More specific comments are listed below:

1. Overall, the manuscript is quite well written in English, however some sentences are a bit strangely phrased. I am not a native English speaker, so I can only give some advice here and suggest revising the whole manuscript accordingly. For example: lines 42-43, I guess the gut microbiota contributes to the protection against toxic compounds and not insects? Lines: 73-75, ...one bacterium in a gut?? Lines 100-105: The sentence is spread over six lines, try breaking it into shorter sentences. Lines 118-120: Rewrite please, difficult to understand the conclusion. Line 122: Microbial biotransformation by *Bacillus* and not of *Bacillus*?. etc...
2. Lines 123-127: Rewrite to state your hypothesis in a way it directly targets *Bacillus* as a group of interest in this study. It should be written that there might be other microbes contributing to the process of biotransformation, however they were not investigated here. You also cite Fig.6 before Fig. 3-5 are actually cited. I think this should be corrected. The same refers to the citation of Fig.3 (line97), I think it goes before Fig. 2 (line 103). Consider revising the whole manuscript accordingly.
3. Line 128: Again, according to Fig. 6, there is little evidence that *Bacillus* would be the dominant microbe in the silkworm gut.
4. Lines 129-130: Is this your conclusion, or a citation of another study? If the latter, please include the reference.
5. Lines 137-138: This cannot be concluded, as other gut microbes were not examined in this study. Also, no real *Bacillus* isolate from the silkworm gut was characterised, neither in vivo, nor in vitro (e.g. using metagenomics or metatranscriptomics). Therefore, we can only speculate here that based on the analysis of previously characterised *Bacillus* sp., silkworm gut *Bacillus* could also contribute to the PIFs biotransformation.
6. Lines 144-146: I would rather say that you evidenced indirectly the contribution of *Bacillus* sp to the toxic compound biotransformation, but we cannot exclude that other microbes can be even more active in this process.
7. Line 160: I would rather say that "...1-5 was rather weak or..", than "greatly weak".
8. Lines 162-164: Not sure we can say that the growth was in good condition, consider rephrasing the sentence.
9. Line 175: 50% survival "rate" and not 50% survival; consider revising accordingly.
10. Paragraph "B. subtilis as a probiotic influenced gut flora by 16S rDNA amplicon sequence analysis", lines 186-206. Have you checked if the bacterium established in the gut? I would be surprised to know that an isolate from a completely different environment than the silkworm gut, well integrates with the insect gut microbiota. Usually, these insect gut microbes have been evolving together with the host over the evolution, are host-specific and often found nowhere else than in this specific gut environment. You could perhaps compare the 16S rRNA sequences of the gut microbiota

and the probiotic strain used, over the time (or at least before, and sometime after the probiotic administration, and also in comparison with the control group)

11. The same paragraph as above; consider revising the header. It reads as if the gut flora was influenced by the 16s rRDA analysis.

12. Not enough details are provided on the 16S rRNA gene amplicon sequencing and data analysis. For example, to say that "DNA samples were amplified" (line:382) is little specific. I would rather give the details on the protocol used, including the information on the primers used to amplify a specific 16S rRNA gene region.

13. Lines 211-217: I think that these two consecutive sentences give the same information, don't they? Please revise to avoid repetition.

14. The same paragraphs (lines 208-239). Is it possible that any component contained in the Bacillus probiotic powder could contribute to this effect? If not, please revise your 16S rRNA data to show any evidence that supplemented Bacillus sp integrated with the gut flora, providing this long-lasting beneficial effect to its host.

The manuscript was carefully revised according to the comments from reviewers. All changes in this manuscript and supplemental material have been highlighted. The changes are detailed in the document of “Point by point response to reviewers”.

For Reviewers' Comments:

For Reviewer 1:

Q1: You should analyse whether the isoflavonoids are taken up by the insect larvae; check if you find them in the integument or body tissues. It could be that a detoxification also occurs inside the insect and not only in its gut microbes.

Response: Thanks for your kind suggestions. We focused the detoxification metabolism by silkworm intestinal microbiota. About the components in the insect tissue, we also tested the distribution by HPLC-MS in supplementary Fig. S218, we found DPL (24) and the corresponding GPIFs were all present in silkworm integument, silk gland, midgut and body fluid, we cannot deny the possibility that the detoxification partially occurred inside the silkworm.

Q2. To evaluate toxicity in cell cultures or insects we need IC_{50} values; I could not find precise values in the MS. Precise IC_{50} values are needed.

Response: Thanks for your kind suggestions. The corresponding data have been supplemented and refined and corrected in the revised manuscript.

Q3. The authors should better consult the existing literature on the chemoeology of plant insect interactions. There are many more relevant mechanisms for an insect to deal with toxic secondary metabolites; discuss the function of ABC transporter, non-adsorption of toxins and rapid passage through the intestines or sequestration and target site inactivation.

Response: Thanks for your kind suggestions. The valuable literature you mentioned have been carefully read and cited in the revised version.

Q4. The presentation and discussion of the microbiome data are inadequate.

Response: Thanks for your kind suggestions. We have further discussed the microbiome data and revised this section in the revised version.

Q5. The English needs improvement by a native speaker.

Response: Thanks for your kind suggestion. The language of this paper has been sent for scientific editing by Springer Nature Author Services. The language editing certificate is attached as an SI file.

For Reviewer 2:

Q1: Overall, the manuscript is quite well written in English, however some sentences are a bit strangely phrased. I am not a native English speaker, so I can only give some advice here and suggest revising the whole manuscript accordingly. For example: lines 42-43, I guess the gut microbiota contributes to the protection against toxic compounds

and not insects? Lines: 73-75, ...one bacterium in a gut?? Lines 100-105: The sentence is spread over six lines, try breaking it into shorter sentences. Lines 118-120: Rewrite please, difficult to understand the conclusion. Line 122: Microbial biotransformation by *Bacillus* and not of *Bacillus*. etc.

Response: Thanks to the comment from you. We have checked the text carefully according to your suggestions and corrected it accordingly.

Q2: Lines 123-127: Rewrite to state your hypothesis in a way it directly targets *Bacillus* as a group of interest in this study. It should be written that there might be other microbes contributing to the process of biotransformation, however they were not investigated here. You also cite Fig.6 before Fig. 3-5 are actually cited. I think this should be corrected. The same refers to the citation of Fig.3 (line 97), I think it goes before Fig. 2 (line 103). Consider revising the whole manuscript accordingly.

Response: Thanks for your kind suggestions. We have isolated and obtained a *Bacillus subtilis* strain and demonstrated that *B. subtilis* existed in silkworm gut and added the related data in the present version. Additionally, we have checked the order of the citation of these figures carefully according to your suggestions and revised them accordingly.

Q3: Line 128: Again, according to Fig. 6, there is little evidence that *Bacillus* would be the dominant microbe in the silkworm gut.

Response: Thanks for your kind suggestions. We have isolated and obtained a *Bacillus subtilis* strain and demonstrated that *B. subtilis* existed in silkworm gut and added the related data in the present version.

Q4: Lines 129-130: Is this your conclusion, or a citation of another study? If the latter, please include the reference.

Response: Thanks for your kind suggestions. In the isolation experiment of silkworm gut microbiota, we found that the *B. subtilis* isolate was as the dominant flora in silkworm gut.

Q5-Q6: Lines 137-138: This cannot be concluded, as other gut microbes were not examined in this study. Also, no real *Bacillus* isolate from the silkworm gut was characterised, neither in vivo, nor in vitro (e.g. using metagenomics or metatranscriptomics). Therefore, we can only speculate here that based on the analysis of previously characterised *Bacillus* sp., silkworm gut *Bacillus* could also contribute to the PIFs biotransformation. Lines 144-146: I would rather say that you evidenced indirectly the contribution of *Bacillus* sp to the toxic compound biotransformation, but we cannot exclude that other microbes can be even more active in this process.

Response: Thanks for your kind suggestions. We took time, isolated and obtained a *Bacillus subtilis* strain, and demonstrated that *B. subtilis* existed in silkworm gut and added the related data in the

present version. We also have demonstrated the conversion capacity of *B. subtilis* strain isolated in vitro (Fig. 4b (iii)).

Q7: Line 160: I would rather say that “..1-5 was rather weak or..”, than “greatly weak”.

Response: Thanks for your kind suggestions. We have corrected it accordingly.

Q8: Lines 162-164: Not sure we can say that the growth was in good condition, consider rephrasing the sentence.

Response: Thanks for your kind suggestions. We have rephrased it accordingly.

Q9: Line 175: 50% survival “rate” and not 50% survival; consider revising accordingly.

Response: Thanks for your kind suggestions. We have corrected it accordingly.

Q10: Paragraph “*B. subtilis* as a probiotic influenced gut flora by 16S rDNA amplicon sequence analysis”, lines 186-206. Have you checked if the bacterium established in the gut? I would be surprised to know that an isolate from a completely different environment than the silkworm gut, well integrates with the insect gut microbiota. Usually, these insect gut microbes have been evolving together with the host over the evolution, are host-specific and often found nowhere else than in this specific gut environment. You could perhaps compare the 16S rRNA sequences of the gut microbiota and the probiotic strain used, over the time (or at least before, and sometime after the probiotic administration, and also in comparison with the control group)

Response: Thanks for your kind suggestions. Here, we have isolated and obtained *Bacillus subtilis* strain and demonstrated that *Bacillus subtilis* existed in silkworm gut in the revised version.

Q11: The same paragraph as above; consider revising the header. It reads as if the gut flora was influenced by the 16s rRDA analysis.

Response: Thanks for your kind suggestions. We have revised it accordingly.

Q12: Not enough details are provided on the 16S rRNA gene amplicon sequencing and data analysis. For example, to say that “DNA samples were amplified” (line:382) is little specific. I would rather give the details on the protocol used, including the information on the primers used to amplify a specific 16S rRNA gene region.

Response: Thanks for your kind suggestions. We have supplemented this section and the details provided on the 16S rDNA gene amplicon sequencing and data analysis in supplementary S7.

Q13: Lines 211-217: I think that these two consecutive sentences give the same information, don't they? Please revise to avoid repetition.

Response: Thanks for your kind suggestions. These two consecutive sentences give different information. In the second sentence, we emphasized the significant effect of *Bacillus subtilis* on body weight of YSY-silkworms. To be more accurate, We have revised it accordingly.

Q14: The same paragraphs (lines 208-239). Is it possible that any component contained in the Bacillus probiotic powder could contribute to this effect? If not, please revise your 16S rRNA data to show any evidence that supplemented Bacillus sp integrated with the gut flora, providing this long-lasting beneficial effect to its host.

Response: Thanks for your kind suggestions. The Bacillus powder was a single bacterium, and no other bacteria were incorporated into the preparation process. We have further discussed the microbiome data and revised this section in the revised version.

Reviewers' comments:

Reviewer #1 (Remarks to the Author):

The revision is adequate.

Reviewer #2 (Remarks to the Author):

Dear Author,

Based on the rebuttal and marked changes in the main manuscript, I could see that the authors have tried to address many of the raised comments. However, some of the comments were not addressed specifically, and the authors have replied with the same answer to many of my comments. I can see the point of the authors insisting on the fact that *Bacillus* is a good probiotic. However, *Bacillus* doesn't seem to be a dominant group in the 16S rRNA study, and other bacteria might also be "hidden" probiotics. Therefore, I don't agree with the comment that *Bacillus* is a key species in the silk gut, as this was not really evidenced in this study. Nevertheless, the study is of good quality and provides interesting information on the silkworm metabolism, that could help improving silk production in a longer term.

Thank you very much for your attention and consideration for our manuscript entitled "The silkworm gut microbiota involved in metabolic detoxification by glucosylation of plant toxins". In order to better answer the questions, we added the relevant experiments of the isolation and identification of silkworm intestinal microbes to the revised article. According to the request of the email, we have carefully addressed the remaining concerns from Reviewer #2 as follows. All changes in this manuscript and supplemental material have been highlighted.

For Reviewers' Comments:

For Reviewer 2:

Q3: Line 128: Again, according to Fig. 6, there is little evidence that *Bacillus* would be the dominant microbe in the silkworm gut.

Response: Thanks for your kind suggestions. Although it was reported in previous studies that *Bacillus* is one of the dominant microbes in silkworm gut, we did not make a conclusion of *Bacillus* as dominant microbe in the silkworm gut by Fig.6. We actually proved the presence of *Bacillus* in silkworm gut by Fig. 6, and further demonstrate the changes of the intestinal microbes of the experimental group compared with the control group after adding *Bacillus subtilis* as a probiotic. In addition, we have demonstrated *Bacillus* in silkworm gut contributed to the PIFs biotransformation in vitro, which suggested that *Bacillus* also played important role in the silkworm gut. To get our point across more accurately, we have revised the description of this section in the manuscript in Line 133, 154-160, 308.

Q5-Q6: Lines 137-138: This cannot be concluded, as other gut microbes were not examined in this study. Also, no real *Bacillus* isolate from the silkworm gut was characterised, neither in vivo, nor in vitro (e.g. using metagenomics or metatranscriptomics). Therefore, we can only speculate here that based on the analysis of previously characterised *Bacillus* sp., silkworm gut *Bacillus* could also contribute to the PIFs biotransformation. Lines 144-146: I would rather say that you evidenced indirectly the contribution of *Bacillus* sp to the toxic compound biotransformation, but we cannot exclude that other microbes can be even more active in this process.

Response: Thanks for your kind suggestions. Firstly, to better reveal the effects of gut bacteria, we have isolated the intestinal bacteria of silkworm by NA medium plate (in Line 411-418) and three isolates were obtained and identified as *Bacillus subtilis*, *Staphylococcus sciuri* and *Enterobacter*

hormaechei strain, respectively. Then, we further studied the microbial transformation in vitro by three strains from silkworm gut of DPL (24). And we also found that the glycosylation ability of *B. subtilis* from silkworm gut for DPL (24) was more potent than that of the other two strains (Supplementary Fig. S219). *B. subtilis* could glycosylate DPL (24) to form 1-5, however, *S. sciuri* and *E. hormaechei* could only partially convert DPL (24) to produce corresponding GPIFs (Supplementary S219). Therefore, it suggested that the formation of abundant GPIFs in SWFs was closely related to silkworm intestinal bacteria. And we can evidence the direct contribution of *Bacillus* sp. with potent glycosylation capacity of DPL (24) in silkworm gut to the toxic compound biotransformation, but we cannot determine whether *Bacillus* sp. would be the most active in this process. All above mentioned was seen in Line 122-130, 154-160.

Q10: Paragraph “*B. subtilis* as a probiotic influenced gut flora by 16S rDNA amplicon sequence analysis”, lines 186-206. Have you checked if the bacterium established in the gut? I would be surprised to know that an isolate from a completely different environment than the silkworm gut, well integrates with the insect gut microbiota. Usually, these insect gut microbes have been evolving together with the host over the evolution, are host-specific and often found nowhere else than in this specific gut environment. You could perhaps compare the 16S rRNA sequences of the gut microbiota and the probiotic strain used, over the time (or at least before, and sometime after the probiotic administration, and also in comparison with the control group)

Response: Thanks for your kind suggestions. Firstly, we proved that *B. subtilis* was the endogenous bacteria in the silkworm gut by the isolation and identification experiment of the silkworm intestinal microbes in Line 122-130. Then, the experimental groups with *B. subtilis* probiotic increased the relative abundance of the genus *Bacillus* in the silkworm gut compared with the control group not adding *B. subtilis* probiotic demonstrated by 16S rDNA amplicon sequence analysis data in Line 209-220. Therefore, the bacterium, *B. subtilis* could establish in the silkworm gut. In addition, although the proportion of intestinal bacteria is relatively stable in long-term evolution, some works still observed alterations in the fecal microbiome composition of probiotics-treated individuals according to the literatures, for instance, (1) Ferrario, C., *et al.* Modulation of fecal *Clostridiales* bacteria and butyrate by probiotic intervention with *Lactobacillus paracasei* DG varies among healthy adults. *J. Nutr.* 144, 1787–1796 (2014). (2) Goossens, D.A., *et al.* The effect of a probiotic drink with *Lactobacillus plantarum* 299v on the bacterial composition in faeces and mucosal biopsies of rectum and ascending colon. *Aliment. Pharmacol. Ther.* 23, 255–263 (2006). (3) Wang,

C., *et al.* Intestinal Microbiota Profiles of Healthy Pre-School and School-Age Children and Effects of Probiotic Supplementation. *Ann. Nutr. Metab.* 67, 257–266 (2015). (4) Martin, F.P., *et al.* Probiotic modulation of symbiotic gut microbial-host metabolic interactions in a humanized microbiome mouse model. *Mol. Syst. Biol.* 4, 157 (2008).

There are also several reports that it is beneficial to add *Bacillus* as probiotics to hosts, like people or animals. For instance, (5) Aljumaah MR, Alkhulaifi MM, Abudabos AM, *et al.* *Bacillus subtilis* PB6 based probiotic supplementation plays a role in the recovery after the necrotic enteritis challenge. *PLoS One* 15, e0232781 (2020). (6) Cao GT, Dai B, Wang KL, *et al.* *Bacillus licheniformis*, a potential probiotic, inhibits obesity by modulating colonic microflora in C57BL/6J mice model. *J Appl Microbiol* 127, 880-888 (2019). (7) Paytuyi-Gallart A, Sanseverino W, Winger AM. Daily intake of probiotic strain *Bacillus subtilis* DE111 supports a healthy microbiome in children attending day-care. *Benef Microbes* 11, 611-620 (2020).

Q12: Not enough details are provided on the 16S rRNA gene amplicon sequencing and data analysis. For example, to say that “DNA samples were amplified” (line:382) is little specific. I would rather give the details on the protocol used, including the information on the primers used to amplify a specific 16S rRNA gene region.

Response: Thanks for your kind suggestions. We have supplemented this section and the details provided on the 16S rDNA gene amplicon sequencing and data analysis in supplementary S7 in Line 450-487.

Q14: The same paragraphs (lines 208-239). Is it possible that any component contained in the *Bacillus* probiotic powder could contribute to this effect? If not, please revise your 16S rRNA data to show any evidence that supplemented *Bacillus* sp integrated with the gut flora, providing this long-lasting beneficial effect to its host.

Response: Thanks for your kind suggestions. The *Bacillus subtilis* powder used in our work was a single strain, and no other bacteria were incorporated into the preparation process. Besides, to better demonstrate that any component contained in the *Bacillus* probiotic powder did not affect the glycosylation, the *Bacillus* powder sterilized at 120°C for 30min was used to biotransformation of DPL (24) *in vitro*, and the formation of corresponding glycosylation products could not be detected by LC-MS. Any component contained in the *Bacillus* probiotic powder had no effect on the experiment of biotransformation *in vitro*, but whether they have an effect *in vivo* has not been studied. In addition, according to our 16S rDNA data, it increased the relative abundance of the genus *Bacillus* in the silkworm gut after adding *B. subtilis* probiotic, which increased from 1.19% to 2.12% in YSY-silkworm, increased from 0.72% to 1.26% in HK2-silkworm (Fig. 6c). Furthermore, it also increased the gut microbial diversity and overall evenness. And we have revised and discussed the microbiome data in the last revised version and this revised version in Line 209-220.